# Direct observation of hot-electron-enhanced thermoelectric effects in silicon nanodevices

Huanyi Xue[1,8], Ruijie Qian[1,2,8], Weikang Lu[1,3,8], Xue Gong[1], Ludi Qin[1], Zhenyang Zhong[1], Zhenghua An ✉[1,3,4,5], Lidong Chen ✉[6] & Wei Lu[2,7] ✉

The study of thermoelectric behaviors in miniatured transistors is of fundamental importance for developing bottom-level thermal management. Recent experimental progress in nanothermetry has enabled studies of the microscopic temperature profiles of nanostructured metals, semiconductors, two-dimensional material, and molecular junctions. However, observations of thermoelectric (such as nonequilibrium Peltier and Thomson) effect in prevailing silicon (Si)—a critical step for on-chip refrigeration using Si itself—have not been addressed so far. Here, we carry out nanothermometric imaging of both electron temperature ($T_e$) and lattice temperature ($T_L$) of a Si nanoconstriction device and find obvious thermoelectric effect in the vicinity of the electron hotspots: When the electrical current passes through the nanoconstriction channel generating electron hotspots (with $T_e$~1500 K being much higher than $T_L$~320 K), prominent thermoelectric effect is directly visualized attributable to the extremely large electron temperature gradient (~1 K/nm). The quantitative measurement shows a distinctive third-power dependence of the observed thermoelectric on the electrical current, which is consistent with the theoretically predicted nonequilibrium thermoelectric effects. Our work suggests that the nonequilibrium hot carriers may be potentially utilized for enhancing the thermoelectric performance and therefore sheds new light on the nanoscale thermal management of post-Moore nanoelectronics.

With the continuously downscaled Si technology, electrons in nano-sized transistors are driven to be far from thermal-equilibrium with the lattice and reach a much higher effective temperature ($T_e$) than the hosting lattice ($T_L$)[1,2]. The energy relaxation of these electron hotspots with an ultrahigh integration density exerts severe heat load to the chips bottlenecking their functions and performance[3–5]. As a result, thermal management constitute an urgent task in post-Moore nanoelectronics[6,7]. On this aim, conventional heat removal approach is mainly based on passive cooling, which relies on the intrinsic thermal conductance of the materials. Unfortunately, the limited thermal conductance of materials (e.g., $\kappa_{Si} \sim 150\,Wm^{-1}K^{-1}$ at 300 K)[8,9] as well as ubiquitous interfacial thermal resistance occupies a significant portion of the total thermal resistance of the modules, making the external cooling (e.g., highly thermal-conductive packaging) less effective. New cooling solutions are therefore highly demanded to overcome the limitation.

[1]State Key Laboratory of Surface Physics and Department of Physics, Institute for Nanoelectronic Devices and Quantum Computing, Fudan University, 200433 Shanghai, People's Republic of China. [2]National Laboratory for Infrared Physics, Shanghai Institute of Technical Physics, Chinese Academy of Sciences, 200083 Shanghai, China. [3]Shanghai Qi Zhi Institute, 41th Floor, AI Tower, No. 701 Yunjin Road, Xuhui District, 200232 Shanghai, China. [4]Yiwu Research Institute of Fudan University, Chengbei Road, 322000 Yiwu City, Zhejiang, China. [5]Zhangjiang Fudan International Innovation Center, Fudan University, 201210 Shanghai, China. [6]State Key Laboratory of High Performance Ceramics and Superfine Microstructure, Shanghai Institute of Ceramics, Chinese Academy of Science, Shanghai, China. [7]School of Physical Science and Technology, ShanghaiTech University, 201210 Shanghai, China. [8]These authors contributed equally: Huanyi Xue, Ruijie Qian, Weikang Lu. ✉e-mail: anzhenghua@fudan.edu.cn; luwei@mail.sitp.ac.cn

In contrast to the passive method, semiconductor thermoelectric (TE) cooling is an active approach, which has attracted much interest in recent years[10–18]. In this approach, heat flow is generated along with the electrical current through the TE effect, namely, $\dot{Q}_{Peltier} = \Pi \cdot I$, where $\Pi$ denotes the Peltier coefficient[19,20]. As a result, the heat removal rate is not restricted by the material property any more. In practice, state-of-art thermoelectrics (TEs) typically adopt two dissimilar materials with different Seebeck coefficients so that the large gradient of $S$ (hence $\Pi$, since $\Pi = S \times T$ according to Thomson relation) at the bi-material interface can generate appreciable cooling power[19–21]. This scheme has turned out to be very successful in macroscopic scale and leads to wide applications in mechanics-free refrigeration. The obtained TE performance is well understood under the hypothesis of local thermal equilibrium (LTE)[22,23]. LTE assumes that $T_e$ and $T_L$ nearly equal to each other ($T_e \simeq T_L \equiv T$) and temperature gradient is typically small enough (small $T$ deviation across large distance) unless $S$ is temperature-dependent resulting in Thomson effect[24]. Under LTE approximation, $\Pi$ is approximately current independent, making $\dot{Q}_{TE}$ exhibit a linear dependence on current[22]. On the other hand, however, LTE approximation for macroscopic TEs may not apply straightforwardly in nanoscale because: (i) electron temperature ($T_e$) can be substantially higher than lattice temperature ($T_L$), as mentioned above, suggesting the conventional definition of $\Pi$ based on the Onsager reciprocal relation may not be valid[25–28]; (2) the highly localized hotspots imply a very high temperature gradient or current density (**J**), which is beyond any practical values possible to occur in macroscopic scale[28–30]. Besides, to realize microscopic (TE) applications, the emergent single-material TEs seems to be superior to the bi-material structure considering both the nanodevice design as well as material compatibility with Si technology.

The TE Peltier effects of semiconductors under nonequilibrium conditions ($T_e \gg T_L$) have been theoretically researched by several methods[23,25,31–33]. For example, Zebarjadi et al.[33] investigated the nonequilibrium TE effect of single parabolic band semiconductors and concluded that $\Pi$ should be redefined to be related to $T_e$ (instead of $T_L$). It was also found that $\Pi$ exhibits the nonlinear dependence on current density[25,31–33], which could largely enhance the cooling performance under a high electric field. The nonlinear features of $\Pi$ can be attributed to the contribution of high-order terms of current: $\Pi^* \propto I^2$ [25,31–33], where $\Pi^*$ is nonlinear part of $\Pi$. Experimentally, only very few works have so far reported the observation of the nonlinear features of $\Pi$ as nanothermometric capability is necessary but remains challenging particularly for $T_e$[34,35]. Until recently, related phenomena were observed in single-material (graphene) nanostructures under high field regime and were explained by the high drift velocity induced electron wind effect[13] or van der Waals barrier enhanced hot carried effect[14]. The direct experimental evidence of electrons being hot is however still lacking and, accordingly, the redefinition and quantitative studies of the Peltier coefficient under the nonequilibrium conditions remain unrealistic.

Here, we study the TE of a single-material, i.e., Si, the dominant semiconductor in state-of-the-art integration circuit technology and investigate the nonlinear Peltier coefficient induced TE effect both experimentally and numerically at nonequilibrium states. The quantitative measurements suggest that the $T_{eg}$ has a significant impact on modulating the Peltier coefficient, leading to an unconventional TE cooling with a pronounced third-power dependence of current. This abnormal feature is attributed to the theoretically predicted nonequilibrium TE effects induced by the extremely large gradient of $T_e$, and is further supported by the experimental evidence. More importantly, numerical simulation results show excellent agreement with experimental findings, and thus give strong evidence that the hot electron can be used as an energy carrier for enhancing the TE performance.

## Results

### Nano-constriction architecture

To realized highly localized hotspot in Si, the nanoconstriction structure is conceived and fabricated in the phosphorus (P)-doped ($\sim 1 \times 10^{19}/\mathrm{cm}^3$) Si films (~90 nm thick) epitaxially grown on high resistivity Si substrate. The width of the narrowest region is about 400 nm so that the electric field is mostly concentrated here under external bias (see Supplementary Fig. 8). Figure 1a is the atomic force microscopy (AFM) image of the height signal around the constriction, displaying the smooth surface of channel and the sharp etching boundaries. The etching depth of about 120 nm as confirmed by AFM is higher than the thickness of the doped film, insuring all the electrical current squeeze within the constricted epitaxial layer. The bias voltage (either sine wave or square wave) is applied between the source and the drain through the ohmic contact electrodes. The linear current-voltage ($I$-$V$) characteristic confirms the good Ohmic contact, which

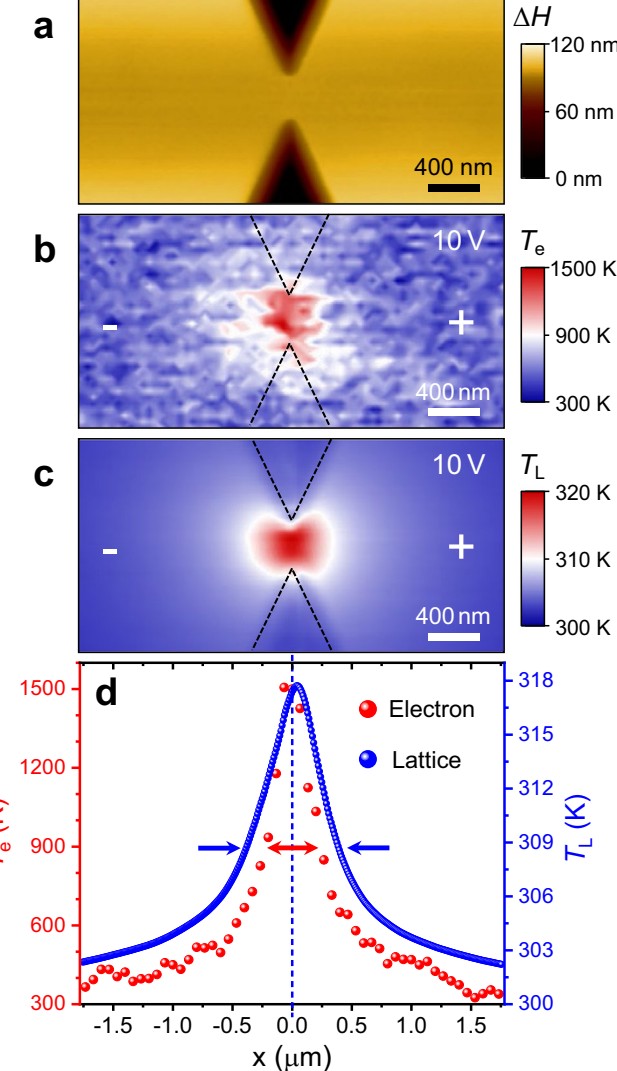

**Fig. 1 | Nanoscale thermal mapping of heated electrons and heated lattice. a** AFM. Topography of Si device, with a 400 nm wide constriction and ~120 nm height variation ($\Delta H$). **b** SNoiM. 2D image of $T_e$ on a narrow conducting channel biased with $V_b = 10$ V. **c** SThM. 2D image of $T_L$ taken on the same device as that for SNoiM with $V_b = 10$ V. **d** 1D profiles of $T_e$ (red dots) and $T_L$ (blue dots) taken along the channel under the same bias condition with **b** and **c**, where the width of $T_e$ and $T_L$ profiles are indicated by the arrows. The dashed vertical blue line marks the center of constriction for guiding the deviation of $T_L$. The polarities of applied voltages are denoted as + and −.

rules out any possible influence from rectifying behavior of contact electrodes to the results shown in this work (see Supplementary Fig. 1).

## Electron and lattice subsystems out-of-thermal-equilibrium

To directly explore the nonequilibrium signature of our device, we adopt a unique methodology that combines scanning noise microscope (SNoiM) with scanning thermal microscopy (SThM) techniques to spatially probe the $T_e$ and $T_L$ from electron/lattice subsystems separately. SNoiM is a recently developed noncontact radiative electronic nanothermometry, which detects near-field fluctuating electromagnetic field at about $20.7\pm1.2$ THz originating from random thermal motions of hot electrons and therefore provides information on $T_e$-distribution. In contrast, SThM is contact nanothermometry which probes the local $T_L$-distribution using a scanning thermocouple tip. Details for the measurement principle and methods of SNoiM/SThM can be found elsewhere[36] (also see "Methods" and Supplementary Note 3 and 4). All the thermometric measurements are carried out at ambient temperature ($T_{Room} \sim 300$ K) and atmospheric pressure. Figure 1b, c display the two-dimensional (2D) spatial distribution of $T_e$ and $T_L$ around the nanostructures under the AC square-wave bias of 10 V with the same scanning area as Fig. 1a. Both the hot spots of $T_e$ and $T_L$ are visible and concentrated at the bottleneck of the conducting channel. More measurements under different biases and on different devices confirm similar behaviors. It is noteworthy to point out that, unlike in GaAs[35,36] where asymmetric hotspot for $T_e$-pattern was observed, no appreciable asymmetry was found here within the biased range (up to 10 V) (also see Supplementary Note 4). This implies that the electron transport here can be viewed as diffusive and local in this dimension, without involving nonlocal energy dissipation mechanism seen in GaAs. For quantitative analyze, the peak electron temperature is evaluated to reach above 1500 K ($\Delta T_e \sim 1200$ K) in the $T_e$-profile (Fig. 1b), whereas under the same bias voltage the maximum value of $T_L$ in the lattice subsystem is only around 320 K ($\Delta T_L \sim 20$ K). This remarkable temperature difference justifies the existence of a nonequilibrium phenomenon between conduction electron and lattice subsystems. For a detailed comparison between $T_e$ and $T_L$, the one-dimensional (1D) thermal profiles of $T_e$ and $T_L$ taken along the channel (under the same bias condition with Fig. 1b, c) are plotted together in Fig. 1d, where the vertical scales of $T_e$ and $T_L$ are adjusted to make the peak heights align. It is obvious that the line shape of $T_L$ is slightly broader than that of $T_e$ (as indicated by the arrows). The sharp and narrow $T_e$-peak exhibits a typical nonequilibrium feature resulting from a relatively small specific heat and low thermal conduction of the electron subsystem (as will be discussed later with numerical simulation). Remarkably, it should be noted that the position of the highest value of $T_L$ deviates slightly away from $T_e$-peak position or center of constricted channel (as marked by the blue guideline). This leads to an asymmetry thermal profile (details for determining the deviation see Supplementary Note 4). In addition, the deviation of peak position shifts to the opposite direction when the bias is reversed, implying that the TE conversion may have readily taken place (see Supplementary Note 4).

## Direct observation of nonequiliubrium TE effect in Si constriction

To spatially resolve the TE effect, a scanning thermal microscope (SThM) equipped with a lock-in technique was performed. In order to extract the pure TE cooling/heating and Joule heating from the total thermal signals detected by tip apex, the alternating voltage with two types of wave modulation (sinusoidal[13–16,37] and square waves[12,38,39]) are applied to the devices. By extracting the first/second harmonic response from square/sinusoidal wave modulation, TE and Joule signals are consequently decoupled to be directly observed (see "Methods" and Supplementary Note 4). Figure 2a–f displays the

spatial resolution of Joule heating (Fig. 2a–c) and TE effect (Fig. 2d–f) around the constriction under various biases. Both the Joule and TE signals are increased monotonously as the voltage rises. To show the evolution of these thermal signals with bias, the color scales are intentionally adjusted. The hot spots of Joule heating are confined within the constriction owing to the high current density of shrink structure. By contrast, the spatial distribution of TE cooling/heating are distributed on both sides of the constriction with the sharp nodes ($\triangle T_{TE} = 0$) in the middle, where the left (right) side display the heating (cooling) signature when current flows along the $+x$ direction. In principle, the TE signals are obtained by multiplying lock-in amplitude $R$ and cosine of phase $\Phi$, thus the sign of TE cooling (negative)/heating (positive) is determined by $\Phi$[38,39]. Figure 2g shows the spatial maps of $\Phi$ under the same bias condition in Fig. 2f, in which $\Phi$ abruptly change -180° at the middle position, and therefore directly demonstrates the sign change of TE cooling/heating. Remarkably, the signal amplitude shown in Fig. 2d–h is much larger than the simple expectation from possible local modification of Seebeck coefficient at the nano-constriction. Therefore, the Seebeck coefficient change cannot play a dominant role for the observed TE results, as will be seen later in Discussion part and in Supplementary Note 7. We mention that when the current direction was reversed, the signals of TE cooling/heating are correspondingly exchanged, suggesting the current-direction dependence of the TE cooling/heating effect[14], as depicted in Fig. 2h. In other words, the hotter (colder) peak due to Peltier effect is always close to the exit (entrance) side of the majority carriers here. This conclusion has been confirmed in multiple devices with similar structure and dimensions. This result is consistent with the observation in nanoconstricted GaAs channel[35,36] but is opposite to the previous report obtained in long nano-crystalline Si microwire channel[40] where minor carriers enhance the Thomson effect and cause lattice overheating within the long channel (entrance side of majority carriers).

Figure 3a–c display the line profiles of the Joule and TE signals under different biasing voltages. The 1D thermal profiles of Joule heating (Fig. 3a) and TE cooling/heating (Fig. 3b) are taken along the constriction under various biases: 4–10 V. Obviously, the intensities of Joule and TE signals increase monotonously as bias rises, while the line shapes of these signals are maintained. The TE cooling/heating $\Delta T_{TE}$ is notably increased up to ~ ±1.3 K with the applied bias of 10 V. Compared to the large amplitude of Joule heating ($\Delta T_{Joule} \sim 18$ K at the bias of 10 V), $\Delta T_{TE}$ is about 7 % of $\Delta T_{Joule}$, which explains that the asymmetry of the total heat (sum of $\Delta T_{Joule}$ and $\Delta T_{TE}$) profile in Fig. 1c is less prominent. The voltage- (current-, owing to the linearity of $I$–$V$ curve) dependent intensities of Joule heating and TE cooling/heating are plotted together in logarithmic coordinates for thorough comparison (Fig. 3c). For quantifying the magnitude of the thermalsignals, Joule signals are defined as the peak values while the TE signals are obtained by averaging the maximum absolute values ($|\Delta T_{TE}|$) at the cooling and heating positions, these values are extracted from the respective line profiles. Joule heating (red dots) exhibits a square dependence on the current, as marked by the fitting curve (light red line), which is in agreement with the Joule–Lenz law: $\Delta T_{Joule} \propto I^2 R$. In conventional TE devices, the thermal gradient is always neglected, thereby according to the definition: $\Pi = S \cdot T$ the Peltier effect is dominated by $\nabla S$, leading to a linear relation between Peltier signals and current: $\Delta T_{Peltier} \propto \nabla \Pi \cdot \mathbf{I} = T \nabla S \cdot \mathbf{I}$ (this linear feature is verified in a semiconductor-metal junction, see Supplementary Note 5). Differently, the Peltier signals observed here are well fitted by a cubic function (light blue line) with current. This anomalous feature is consistent with the theoretical nonlinear Peltier effect that predicted appear in nonequilibrium state. More importantly, owing to the third power of current in TE signals, the ratio of Peltier to Joule signals is increased linearly with bias voltage (see inset in Fig. 3c).

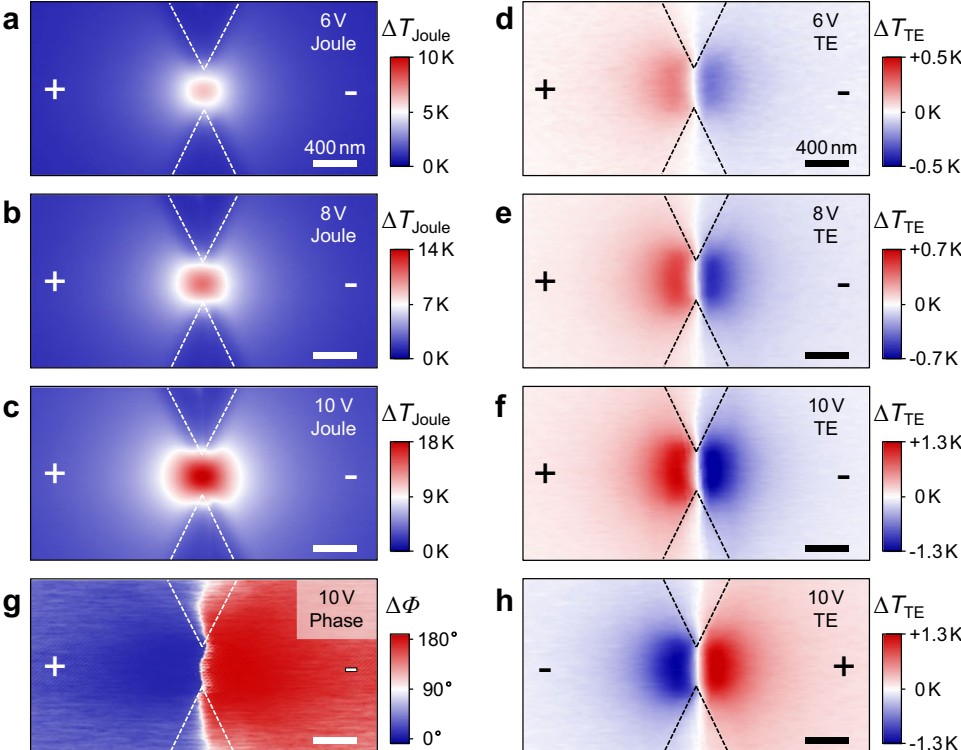

**Fig. 2 | Decoupling detection of Joule and TE signals from TL by SThM. a–c** 2D images of Joule heating under AC voltages (sinusoidal wave) varying from 6 to 10 V, which are demodulated at the second harmonic. **d–f** 2D images of thermoelectric (TE) cooling/heating under AC voltages (square wave) varying from 6 to 10 V with same polarity, and are demodulated at the first harmonic (see Supplementary Note 4). **g** Phase ($\Phi$) image of TE signal under the same bias condition with **f** directly shows the abrupt 180° change at the middle of constriction. **h** 2D image of TE cooling/heating at opposite bias polarity of **f**. The color scales of Joule and TE maps are intentionally adjusted to show the evolution of signals with the voltages. All scale bars are 400 nm.

## Discussion

In order to understand the TE transport mechanisms in our devices, we start from the general formulations for thermal transport in current-carrying structures involving TE effects. To make practical formalisms we make the following assumptions that: (i) electron and lattice sub-systems are decouplable with their mutual energy exchange determined by electron-phonon scattering, and hence a two-temperature model is applicable in the nonequilibrium TE transport; (ii) the thermal transport within each subsystem follows the classical electrical and thermal transport laws, in particular, the non-Fourier heat transport in nanoscale is taken into account with a modified (effective) thermal conductivity; (iii) the heat diffusion in each subsystem is isotropic and the involved phonon dynamics of different phonon modes are not considered for simplicity. Under these assumptions, the thermal transport of electron subsystem in steady-state conditions can be written as,

$$\nabla \cdot (\kappa_e \nabla T_e) + \nabla \cdot (\kappa_L \nabla T_L) + \frac{J^2}{\sigma} - \nabla \cdot (\Pi \mathbf{J}) = 0 \qquad (1)$$

Where $\kappa_e$, $\kappa_L$ are the thermal conductivity within the electron and lattice subsystem, and $\sigma$ is the electrical conductivity. This formula can be reduced to the usual quasi-equilibrium case when $T_e$ approaches $T_L$. On the other hand, it should be stressed that the Peltier coefficient $\Pi$ in this general form of Eq. (1) should take account in the nonequilibrium contribution. The TE signal in the experiments arises from the term of $-\nabla \cdot (\Pi \mathbf{J})$ which can therefore be termed as Peltier effect. However, phenomenologically the observed signal appears like Thomson effect as the large temperature gradient ($\nabla T_e$) is needed (See Supplementary Note 8).

To evaluate the nonequilibrium Peltier coefficient, we notify that, under the bias condition the local average kinetic energy of electrons $\langle E_k(r) \rangle$ can be divided into the drift part $\frac{P_0^2}{2m^*}$ and the diffusion part $\frac{(P-P_0)^2}{2m^*}$, where $P$, $P_0$ and $m^*$ are denoted the local momentum, collective momentum and effective mass of the electron. The drift part reflects the collective motion of electrons caused by the Fermi sphere shift, and the diffusion part represents the isotropic motion relative to collective velocity defined by the fluctuation around the Fermi surface. According to the theoretical calculations[33], the local nonequilibrium Peltier coefficient here is simplified as

$$\Pi(r) = \frac{1}{e}\left( -\mu(r) + \frac{P_0^2}{2m^*} + \frac{5}{3}\frac{P^2}{2m^*} \right) \qquad (2)$$

here, $\mu(r)$ is the local quasi-Fermi level. Considering the high doping concentration ($\sim 1 \times 10^{19}/cm^3$) of the device, $\mu(r)$ and $\frac{P^2}{2m^*}$ are approximated in the degenerate limit, where $\frac{P^2}{2m^*} \approx \frac{3}{5}\mu(r) + \frac{3\pi^2}{10}\frac{(k_B T_e)^2}{\mu(r)}$ and $\mu(r) \approx \frac{3}{2} k_B T_e$[41]. Moreover, due to the neglected contribution of the drift part to total kinetic energy, $\frac{P_0^2}{2m^*}$ is omitted as a reasonable approximation. Hence, we have $\Pi(r) \approx \frac{\pi^2}{3e} k_B T_e$ which directly shows that under the nonequilibrium condition the Peltier coefficient is governed by $T_e$, thus allowing us to simulate the nonequilibrium TE effect by using the distribution of $T_e$ that extracted from the two-temperature model (see Supplementary Note 6).

In order to obtain the spatially dependent electron temperature profile $T_e(r)$, we note that the Joule heat term and the electron-phonon scattering term are respectively the dominant heat generation and

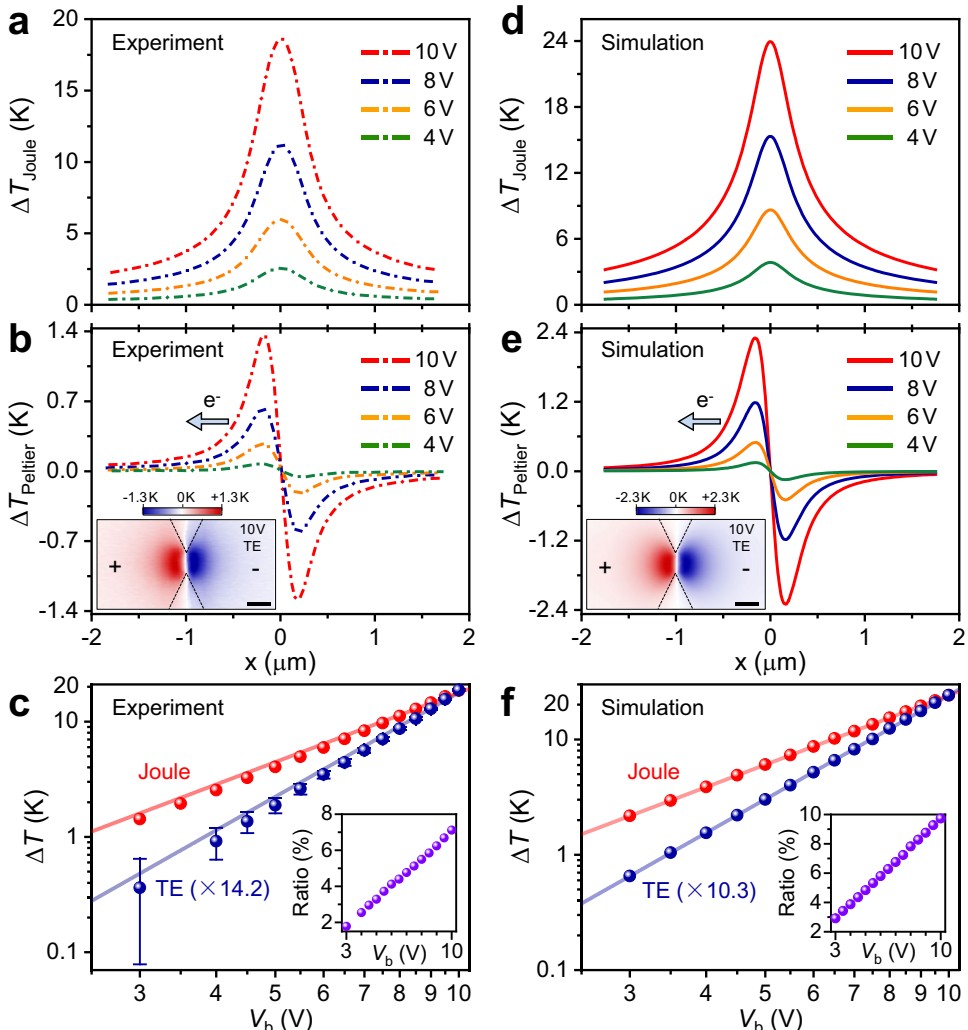

**Fig. 3 | Comparison between experimental data and simulation results.**
**a**, **b** Experimental 1D profiles of Joule and TE signals at the $V_b$ of 3–10 V. The 2D TE map shown in **2f** is reproduced as inset in **b** for comparting with the simulated results. **c** Voltage (current) dependence of Joule (red dots) and TE (blue dots) signals with quadratic (light red line) and cubic (light blue line) fitting. Error bars are 20-mK of $\triangle T_{TE}$ uncertainty as extracted from the standard deviation of multiple

SThM measurements. The inset shows the evolution of the ratio of TE to Joules with voltage (purple dots). **d**, **e** Simulated 1D profiles of Joule and TE signals at the $V_b$ of 3–10 V. The inset in **e** is TE map ($V_b = 10$ V) extracted from the simulations. **f** Voltage (current) dependence of simulated Joule (red dots) and TE (blue dots) signals with quadratic (light red line) and cubic (light blue line) fitting. The inset shows the evolution of the ratio of TE to Joules with voltage (purple dots).

heat dissipation mechanisms, while the TE term is a small correction to the total heat distribution, as suggested also by the experimental data (Figs. 2 and 3). Hence, $T_e(r)$ profile is determined mainly by the balance between these two dominant terms. Using finite element method, we have performed three-dimensional modeling with Eq. (1) by neglecting the TE term as a first order approximation. $T_e(r)$ and $T_L(r)$ profiles are therefore extracted. The simulated results under DC bias of 10 V display similar nonequilibrium features to experiments with the analogous peak values ($T_e \sim 1596$K and $T_L \sim 325$K) and line features (see Fig. 4b and Supplementary Fig. S9). For further comparing the simulated results with experimental data, the simulated $T_e$ profiles with increasing bias from 4 to 10 V (color lines) are plotted together with experiment results (color dots) in Fig. 4b, where both the peak values and line shapes are in excellent agreement with the experimental data, hence, assuring the reliability of the numerical method. With the $T_e(r)$ profile known, $\Pi(r)$ distribution is known and can be substituted into the total thermal transport equation, i.e., Eq. (1) and Joule heating and TE cooling/heating can be computed.

The simulated results of Joule heating and TE cooling/heating are shown in Fig. 3d–f for direct comparison with experimental data. The

simulated thermal profiles of Joule (Fig. 3d) and TE (Fig. 3e) thermal profiles exhibit the same features as in experiments, including the line shapes, peak values, the distance between cooling and heating peaks and evolutions with bias increasing. Furthermore, the simulated results for 2D TE distribution at 10 V bias (inset of Fig. 3e) also display the analogous contour profiles and TE signs to those of the experiment (inset of Fig. 3b). Interestingly, the simulated Joule and TE (Fig. 3f) clearly show the square and cubic dependence to bias (current) respectively, thus causing a linear increase of the ratio of TE to Joule signals, consistent with experimental results.

To consider the underlying microscopic mechanism behind the above thermoelectric transport, Joule heating and TE cooling/heating are analyzed separately. As presented in Fig. 4a, the conductive electrons are excited by the high electric field when passing through the constriction and consequently causing a nonequilibrium region with an extremely large gradient of $T_e$ profile. During the whole process, the excited electrons (hot electrons) in electron system would release energy to the lattice system via the electron-phonon scattering (corresponding to symmetric Joule heating). In addition, when the $T_e$ modulated $\Pi$ profile is given, hot electrons play as energy carriers that

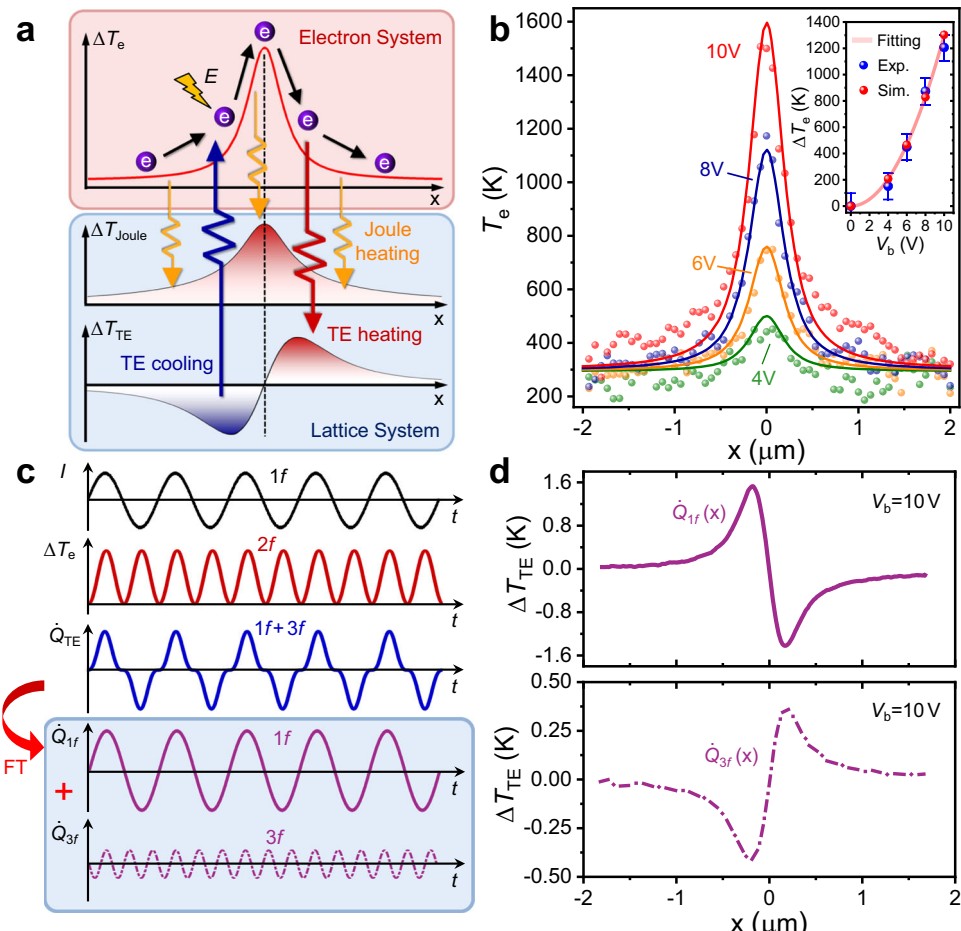

**Fig. 4 | Signature for Te modulations in nonequilibrium TE effect. a** Schematic illustration of energy flow within/between electron (top panel) and lattice systems (bottom panel). **b** 1D profiles of experimental measured $T_e$ (dots) and simulated (solid lines) $T_e$ taken along the channel at various $V_b$ of 4–10 V. The inset shows the voltage (current) dependent peak values of $\Delta T_e$ ($T_e - T_{Room}$) extracted from experiments (blue dots) and simulations (red dots), which matched well with quadratic fitting ($\propto I^2$) as indicated by the light red line. Error bars in inset are 100-K of $T_e$-fluctuation as extracted from the noise level of SNoiM measurement. **c** When a

sinusoidal wave current $I$ with a frequency of $f$ is supplied to the device, the $\Delta T_e$ ($\propto \nabla \Pi$) is then oscillated by the frequency of $2f$, hence owing to the correlation: $\dot{Q}_{TE} \propto \nabla \Pi \cdot \mathbf{J} \propto I^3$, the total TE signals ($\dot{Q}_{TE}$) would be modulated by the admixture of first ($\dot{Q}_{1f}$) and third ($\dot{Q}_{3f}$) harmonic, which can be decoupled by flourier transform (FT). **d** Measured 1D profiles of $\dot{Q}_{1f}(x)$ (top panel) and $\dot{Q}_{3f}(x)$ (bottom panel) along the channel at $V_b = 10$ V, which are demodulated at the first and third harmonic by the lock-in technique.

absorb energy from lattice (Peltier cooling) as they climb the $T_e$ profile at the entrance side, and release excess energy to lattice (Peltier heating) on their way down.

The observed unconventional cubic dependence of TE cooling/heating with current is attributed to the contribution of $T_e$ in nonequilibrium Peltier coefficient, owing to the height of $T_e$ profiles ($\propto \Delta T_e$) are proportional to the square of the current (as shown in the inset of Fig. 4b), which is examined by an additional experiment. To further reveal this $T_e$ modulated TE effect, here we use a sinusoidal wave modulated voltage (current). As shown in Fig. 4c, the applied current $I$ is fluctuated with time $t$ by a constant frequency f, as: $I\sin(\omega t)$, here $\omega$ is angular frequency ($\omega = 2\pi \cdot f$). As mentioned above $\Delta T_e \propto I^2$, hence $\Delta T_e \propto \sin^2(\omega t)$. Following, the current dependence of TE cooling/heating signal is $\Delta T_{TE} \propto \dot{Q}_{TE} \propto \nabla \cdot (\Pi \mathbf{J}) = \nabla \Pi \cdot \mathbf{J} \propto \nabla T_e \cdot \mathbf{J} \propto \Delta T_e \times I \propto I^3 \propto \sin^3(\omega t)$, clearly showing the origin of the nonlinear Peltier. Moreover, by Fourier transform (FT), $\Delta T_{TE}$ is decomposed into parts: $\Delta T_{TE} \propto 3\sin(\omega t) - \sin(3\omega t)$, which can be detected by lock-in technique respectively at the fundamental and third harmonic frequencies. It should be noted that the magnitude of the first part (fundamental frequency, $f$) is 3 times larger than that of the second part (third harmonic frequency, $3f$), in addition the second part possess an additional phase of 180° to current, leading to a sign change compared

with the first part. These features are successfully reproduced in experimental results, as shown in Fig. 4d, both the line profiles of $1f$ and $3f$ component of thermal signals exhibit the analogous shape features with total TE (as shown in Fig. 3b and e), but the line profile of $3f$ component shows a opposite line shape to $1f$, in addition the peak value of $1f$ is about 3-4 times larger than that of $3f$. Whereas in contrast to the nonequilibrium TE effect, the TE Peltier signals under LTE condition at Cr-Si junctions is not show the same features in $3f$ component (see Supplementary Fig. S6), further demonstrating that it is an intrinsic property of the nonequilibrium TE effect.

Finally, we mention several other mechanisms which may play certain role in the TE transport of our devices. First, the energy filtering effect[42,43] and the thermionic emission cooling[15,44] requires a potential barrier at the interface of dissimilar materials, hence cannot contribute in our device as the channel is the single material. Second, the boundary scattering to electron transport may alter the effective Seebeck coefficient and may play a significant role for devices with channel size being smaller than or comparable to the characteristic mean free path of electrons, as discussed in ref. 13. In our devices, however, the width of the nanoconstricted neck (~400 nm) and the size of the hotspots (~300 nm) are substantially larger than the mean free path of

transporting electrons (less than 20 nm)[45,46] in the room-temperature devices and therefore the boundary scattering effect cannot be a dominant mechanism, which is also supported by the active heated-probe local thermovoltage measurements[13,47] (see Supplementary Note 9). Third, the conventional Thomson effect under quasi-equilibrium conditions ($T_e \cong T_L$) relies on the local temperature and the temperature gradient of Seebeck coefficient, which is characterized by the Thomson coefficient $\beta = T \cdot \frac{dS(T)}{dT}$. In our work, however, the lattice temperature rises only by a very small amount ($\Delta T_L \sim 20\,K$) and the Seebeck coefficient for P-doped Si is not change significantly around 300 K[48]. Using the typical value of Thomson coefficient of Si, the TE cooling due to this effect is estimated being only ~4% of the measured values (see Supplementary Note 7). Fourth, considering the detailed phonon dynamics being involved in the energy transport, directional phonon emission[36] during the electron-phonon scattering may further enhance the observed phenomena here. However, it remains to be an interesting open question how the third-power dependence of the observed temperature difference on the electrical current can be preserved if such a mechanism plays a considerable role and this certainly requires further studies from theorists.

In summarizing, by combining the SThM and SNoiM techniques, the temperature of electron and lattice in Si nanodevices are allowed to be investigated separately. We observe the coexistence of geometry-dependent TE signals and the strong nonequilibrium effect in P-doped Si nanoconstrictions. Under the nonequilibrium condition, according to the theoretical prediction, we defined the Peltier coefficient by using the electron temperature, which allows us to numerically analyze the TE effect. Based on this method, the simulation results are in excellent agreement with the experimental data, strongly confirming the evidence for the existence of nonequilibrium TE effects and demonstrating the accuracy of nonlinear $\Pi$ definition. It should be noted that, although the intensity of TE signals is comparatively smaller than Joule heating, the contribution from $T_e$ in the Peltier coefficient results in the nonlinear behavior, leading to the remarkable enhancement with rising bias, which is far different from with conventional TE principle.

This work experimentally demonstrates the approach for enhancing the TE effect by electron temperature under nonequilibrium conditions, which may be potentially applied for future on-chip cooling. A wide range of different methods may be considered to further enhance the TE property, such as laser-spot-induced temperature gradient[49], intervalley-assisted energy transport[36], spin-Peltier effect[50] etc.

## Methods

### Nanodevice fabrication
The nano-constriction are patterned on phosphorus-doped (n-type) Si films (~90 nm thick) deposited on intrinsic Si (001) wafer by epitaxial growth method.

### SNoiM
A home-built Scanning Noise Microscope (SNoiM) is used for studying the electron temperature. SNoiM is so far the only instrument that visualizes hot electrons in the steady-state transport condition. SNoiM exclusively senses evanescent radiation localized on the material surface, but does not sense the familiar THz photon emissions such as those due to the blackbody radiation[51], externally induced coherent electron motion[52], and the one-particle radiative transition between the initial and the final states[53]. This is because all those photon emissions do not yield evanescent field on the material surface. Detected with SNoiM is the charge/current fluctuation that generates intense evanescent waves but cancels out in the region away from the surface. In this work it is the hot-electron shot noise, the intensity of which is most simply

characterized by the effective electron temperature $T_e$. Absolute values of $T_e$ are derived from the signal intensity without using any adjustable parameter. The spatial resolution determined by the size of tungsten tip is about 50 nm. The principle and the construction of SNoiM are described in more details in refs. 35,36 and in Supplementary Note 3.

During the experiments of Si nano-constriction device, an external pulsed bias is applied and the signal is demodulated with $f_{bias}$=5 Hz. Absolute values of $T_e$ are derived from the signal intensity measured under different bias voltage (see Supplementary Note 3).

### SThM
A commercial Scanning Thermal Microscopy (SThM) based on Bruker Dimension Icon atomic force microscope (AFM) is used to map the lattice temperature distribution. A nanoscale thermocouple-type probe technology was adopted for measuring the sample lattice temperature, similar to previous work[54]. The thermo-voltage of the thermocouple is sent to the preamplifier and then to the imaging amplifier (~1 nV@1 kHz bandwidth and 1000× gain). The amplified thermal signal was input into the AFM controller for data acquisition using the commercial AFM software. The system is working in the ambient environment and the spatial resolution was determined by the tip size of the thermocouple (about 50 nm). To calibrate the temperature sensitivity of the tip, we measured the SThM signal on Si sample heated to a known temperature and obtain the sensitivity of the probe to be 13 μV/K. Hence the SThM transfer characteristic can be established linking its signal to the local temperature of the sample under study. We mentioned that the temperature measured with SThM is the lattice temperature $T_L$ but less affected by the hot electrons, because heat flow is dominated by the lattice that has a heat capacity several orders of magnitude larger than that of conduction electrons. In the experiment, the device is excited by an AC or pulsed voltage through the global ohmic contacts at a frequency of $f_{exc}$ = 413 Hz. During the SThM measurement, the tip is scanned over the sample with contact mode, and the thermal voltage collected by the tip is sent to the VertiSense imaging amplifier and demodulated by a SRS830 lock-in amplifier. More details of our SThM measurements can be found in Supplementary Note 4 (passive method) and Note 9 (active method).

## Data availability
The data that support the findings of this study are available within the article and its Supplementary Information. Additional relevant data are available from the corresponding authors upon request.

## Code availability
The source code to run all analyses in this paper can be obtained from the corresponding authors upon request.

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

## Acknowledgements

Z.A. and W.L. acknowledge financial support from the National Natural Science Foundation of China (NSFC) under grant Nos. 12027805, 11991060, 61521005, 11634012, and 11674070; the Shanghai Science and Technology Committee under grant Nos. 18JC1420400,

18JC1410300, 20JC1414700 and 20DZ1100604; the Sino-German Center for Research Promotion (No. M-0174).

## Author contributions

H.X. and Z.A. conceived the idea and designed the experiments. R.Q. collected the SNoiM data presented in this work. H.X., X.G., and L.Q. performed the passive SThM measurements. H.X. developed and performed the numerical simulations. W.K.L. performed the active SThM measurement. Z.Z. grew the P-doped Si single crystal thin films. H.X., L.C., and Z.A. performed the data analysis. H.X. and Z.A. co-wrote the manuscript with comments from all authors. Z.A. and W.L. co-supervised the project.

## Competing interests

The authors declare no competing interests.
