## [Peer Review File · Nature Communications]

REVIEWER COMMENTS

Reviewer #1 (Remarks to the Author):

The paper by Xue et al "Direct Observation of Hot-Electron-Enhanced Peltier Effects in Silicon Nanodevices" reports interesting thermoelectric phenomena in Si sub-micron nano-constriction that may have significant impact in the field of microelectronics as well as uncovering fundamental thermophysical phenomena. While clearly novel results are presented, the paper has significant omissions and gaps both in the description and explanation of these results as well as insufficient proof of reliability of the methodology used. Unless these are properly addressed requiring, in my opinion, a major revision of the manuscript with, possibly, additional experimental verifications, it cannot be recommended for the publication in the Nature Comms.

Major points.

1. While the discussion makes it clear that the origin of the observed nonlinear Peltier effect in the Si nanoconstriction is the heating of the electron subsystem in the junction – with the constant Seebeck coefficient simply due to Peltier coefficient dependence on the temperature $\Pi = S \times T$ (Surprisingly, this relation was not directly mentioned in the manuscript), the possible local modification of the Seebeck coefficient has to be ruled out. The origin of this effect has to be mentioned in the Results section pointing to the details in the Discussion section.
2. The results such as in the Figure 2 d-h and Fig 3 b,e may also suggest the different Seebeck coefficient in the constriction. In any case, the dependence of the Peltier effect heating for the smaller voltages $\rightarrow 0$ has to be provided in the Fig.3c,f for the completeness.
3. Significantly, in order to rule out the local modification of the Seebeck coefficient in the junction, the scanning thermal gate microscopy (STGM) can be used [Harzheim et al, 2D Materials 2020, 7 (4), 041004. DOI: 10.1088/2053-1583/aba333.] where the hot tip creates the heat gradient and the resulting potential is measured. Such study should be pretty straightforward in the setup.
4. The description of the Scanning Thermal Microscopy to map lattice temperature is very inadequate. The design of the local temperature measurements, that are strongly dependent on the tip-surface contact thermal resistance, the verification of the measured surface temperatures had to be done via one of the existing methods for active or passive devices [see e.g. Menges et al, Nature Comm 2016, 7, 10874, Article. DOI: 10.1038/ncomms10874] and [Evangelini et al, Carbon 2021, 178, 632-639. DOI: <https://doi.org/10.1016/j.carbon.2020.12.012>].
5. The experimental setup details of the SThM and error analysis are also very insufficient. Without these, the validity of the results cannot be verified.
6. While the description of the SNoiM is reasonable in the references 35, 36 a description of this method with the experimental setup used has to be presented in the SI and summarised in the Methods section.

Minor points.

1. The English of the manuscript, while reasonable, has to be carefully revised by the native speaker.
2. Error bars has to be provided or indicated in the graphs.
3. The heat transport Eq. 1 relates to both electron and lattice that has to be corrected.

Reviewer #2 (Remarks to the Author):

The authors report the direct observation of the charge-to-heat current conversion in a Si-based nano-constriction by means of SNoiM and SThM techniques. Thermoelectric cooling/heating signals were observed to appear in the nano-constriction despite the absence of junctions comprising different materials. The authors attribute the observed behavior to the non-equilibrium between electron temperature T_e and lattice temperature T_l . Active thermal management for Si-based electronic devices is of crucial importance, and this work provides a fundamental step to realize such a functionality. The manuscript is well written and organized. However, for further consideration, the following issues must be addressed.

1) The observed phenomenon should not be called the Peltier effect but is rather closer to the Thomson effect. The temperature change observed in the nano-constriction device is caused by ∇S , which is indeed the origin of the Thomson effect. I understand that the observed phenomenon due to the difference between T_e and T_l is different from the conventional Thomson effect. Nevertheless, the non-linear thermoelectric phenomenon due to ∇S should be phenomenologically categorized into the Thomson effect, not the Peltier effect. The authors should reconsider and discuss this point.

2) The quantitative accuracy of the T_e measurements using SNoiM should be demonstrated. For example, at local thermal equilibrium conditions, T_e should be the same as T_l . Thus, SNoiM data should exhibit T_e distributions comparable to T_l distribution detected by SThM. Such an experimental demonstration under local thermal equilibrium conditions should be provided as a control experiment.

3) The authors should describe how to determine the phase offset in Fig. 2. In the nano-constriction used in this study, the temporal response of the temperature change is affected not only by the device design but also the heat transfer to substrates. Thus, the temperature change should have a delay from the applied charge current, leading to non-zero phase values. However, in the data in Fig. 2, the thermoelectric signals have only 0 degree and 180 degrees without any time delay.

4) What is the periodic pattern on the right side of the phase image in Fig. 2g?

5) Related to the comment 3), the sentence "In principle, the Peltier signals are obtained by multiplying lock-in amplitude R and phase ϕ " (page 5) is not correct. The thermoelectric signals detected by lock-in techniques are often discussed in terms of $R \cdot \cos \phi$, which includes both the magnitude and sign information. However, this factor can be used only when the time delay due to thermal diffusion is negligibly small. Since such a time delay cannot be neglected in the present device, the above sentence is not valid (and "cosine" is missing).

6) The combination of SThM and lock-in detection is one of the key techniques in this study. Thus, the procedures of the measurement conditions should be described in more detail. In the case of the well-

known lock-in thermography, the lock-in frequency must be much smaller than the frame rate of the infrared camera. In a similar manner, in the lock-in SThM, does the lock-in frequency need to be much smaller than the cantilever scanning frequency? Conversely, can the lock-in frequency be much larger than the scanning frequency? I could not find the information on the scanning frequency in the SThM measurements.

7) In the experimental results reported here, the thermoelectric cooling is much smaller than Joule heating. However, as demonstrated in Ref. 38 and Phys. Rev. Applied 11, 034022 (2019), the spatial distribution and temporal response of the temperature change induced by dipolar heat sources (a symmetric pair of heating and cooling sources due to thermoelectric effects) are different from those by single heat sources, such as Joule heating. Thus, in principle, the relative contribution of the thermoelectric effects can be larger by changing the design of the nano-constriction. This point should be discussed.

8) In the thermoelectric measurements, the authors applied a square-modulated AC charge current to the device. Thus, when extracting the first harmonic response, the factor $4/\pi$, i.e., the amplitude difference between the square wave and first harmonic sinusoidal wave, should be taken into account. Is this factor included in the analysis?

9) In Supplementary Note 4, "marked by the blue dashed line in Fig. S4(b)" should be replaced with "marked by the black dashed line in Fig. S4(b)".

Reviewer #3 (Remarks to the Author):

In their paper Xue et al. show that a local non-equilibrium state where the electronic system is hotter than the lattice might alter the local Peltier coefficient. The experiments and their interpretation are very interesting. I would in principle suggest this work for publication, but I have a few small questions.

I have some fundamental understanding issue. In my opinion, the Peltier effect (as well as the Seebeck effect) are happening in the electronic system. Meaning: a difference in electron temperature drives an electrical current (Seebeck effect) and a charge current drives a heat current via energetic electrons (Peltier effect). The reason that the lattice temperature is often used in literature is due to the fact that thermalization between electronic and lattice systems is usually very fast (high electron-phonon interaction strength at room temperature). Do you exclude that Peltier cooling should occur in the electronic system as well? The way I understand your Figure 4a is that you say that the Peltier effect is a consequence of heat transfer between the lattice and the electronic system, which can be either positive (cooling) or negative (heating) depending on the gradient of T_e . But wouldn't you expect already a heating/cooling of the electronic system only due to the Peltier effect (=electron-hole asymmetry of the DOS at the Fermi energy, asymmetry of heat transfer, net heat flow, ...). And would you be able to measure this using your SNoiM method (similar to what you do with SThM, by applying an AC signal to the device and by locking to this signal)?

I am also not sure if I fully understand your explanation of the local variation of the Peltier coefficient. Is it correct that you say that the local change in Peltier coefficient is in your case a non-equilibrium effect which is purely caused by a raise of the electronic temperature ($T_e \gg T_L$). This local variation in electron temperature gives you a spatial variation of the quasi-Fermi level and thus a variation of the Peltier coefficient?

So, out of curiosity, if you present evidence that there is a big difference between the electron and the lattice temperature when Joule-heating your junctions: 1) is this an effect that you would only see for high electric fields? And is there a threshold voltage where this occurs? 2) Since $T_e \gg T_L$ is a non-equilibrium situation, on what time-scale does the system relax/thermalize ($T_e = T_L$)? My feeling tells me that electron-phonon interactions at room temperature are rather strong, so I would guess that this thermalization happens in the nano-second range. Is this within your experimental range?

Minor comment: Figure 2 caption: scale bar 400nm not um.

Reviewer #1 (Remarks to the Author):

The paper by Xue et al “Direct Observation of Hot-Electron-Enhanced Peltier Effects in Silicon Nanodevices” reports interesting thermoelectric phenomena in Si sub-micron nano-constriction that may have significant impact in the field of microelectronics as well as uncovering fundamental thermophysical phenomena. While clearly novel results are presented, the paper has significant omissions and gaps both in the description and explanation of these results as well as insufficient proof of reliability of the methodology used. Unless these are properly addressed requiring, in my opinion, a major revision of the manuscript with, possibly, additional experimental verifications, it cannot be recommended for the publication in the Nature Comms.

Reply: We thank the Reviewer very much for evaluating our result to be “*interesting thermoelectric phenomena*” and for giving the possible comments “...may have *significant impact* in the field of microelectronics as well as uncovering *fundamental* thermophysical phenomena...”. We are also very much grateful to the critical comments which are valuable for us to improve our manuscript. We have addressed each point of the report and tried our best to improve our manuscript accordingly.

Major points.

1. While the discussion makes it clear that the origin of the observed nonlinear Peltier effect in the Si nanoconstriction is the heating of the electron subsystem in the junction – with the constant Seebeck coefficient simply due to Peltier coefficient dependence on the temperature $\Pi = S \times T$ (Surprisingly, this relation was not directly mentioned in the manuscript), the possible local modification of the Seebeck coefficient has to be ruled out. The origin of this effect has to be mentioned in the Results section pointing to the details in the Discussion section.

Reply: We thank the Reviewer for this comment which make us realize that our original description was not clear enough. Following this comment, we have improved the manuscript: (1) introduce explicitly Thomson relation, i.e., $\Pi = S \times T$, in the introduction part on Page 2 of our revised manuscript. (2) mention in the Results section that “...Remarkably, the signal amplitude shown in Figs. 2d-h is much larger than the simple expectation from possible local modification of Seebeck coefficient at the nano-constriction. Therefore the Seebeck coefficient change cannot play a dominant role for the observed thermoelectric results ...” and meanwhile point to the details in Discussion section and Supplementary Note 7. (3) in Supplementary Note 7, add the expected signal derived from the possible local modification of the Seebeck coefficient which gives only a very small fraction of the signal (~4% of the measured data), as shown also in Fig. R1.

Fig. R1(a) Reasonable local Seebeck coefficient change due to temperature rise at the nanon-constriction (Data taken from Hinsche et al and Stranz et al (Ref. 48)). (b) Simulated lattice temperature profiles for +10V and -10V bias voltages. The deviation is dramatically smaller than our experimental data (Figs. 2d-h, Fig.3b). (c) Derived Thomson signal profile with the amplitude of the asymmetric curve being only ~4 % of the real observed signal (Fig. 3b).

2. The results such as in the Figure 2 d-h and Fig 3 b,e may also suggest the different Seebeck coefficient in the constriction. In any case, the dependence of the Peltier effect heating for the smaller voltages $\rightarrow 0$ has to be provided in the Fig.3c,f for the completeness.

Reply: We thank the Reviewer for this comment and suggestion and the answer partly overlaps our response to the last question. Indeed, the immediate reason which we can come up with for Figs. 2d-h and Figs. 3b,e is the different Seebeck coefficient in the constriction region. However, after quantitatively evaluating the possible change of Seebeck coefficient due to several possible mechanisms, we conclude that the local change of Seebeck coefficient cannot be the dominant reason for the observed signal amplitude in Figs. 2d-h and Figs. 3b,e, as we discuss in detail in the Discussion part. We have added the above Fig. R1 as Fig. S10 in the newly added Supplementary Note 7. The two most likely mechanisms to induce local Seebeck coefficient change are: (1) the geometrical confinement as been reported in graphene nano-constricted channel in Ref. 13. However, the characteristic mean free path of transporting electrons (less than 20 nm) is negligibly small compared to the width of the nanoconstricted neck (~400 nm) and hence the geometrical confinement effect cannot be so strong to cause large thermoelectric signal. Experimentally, no discernible Seebeck coefficient change is confirmed by additional experiments with heated tip (see Fig. S12 h). (2) Joule heating induces the temperature rise at the constricted neck which thereby alters the local Seebeck coefficient due to its temperature dependence (Thomson effect). According to typical Thomson coefficient values from Si with similar doping level ($\sim 0.47 \mu\text{V}/\text{K}^2$), the simulated signal amplitude is only ~4 % of the real measured one. Therefore, in order to produce such a large signal as in experiments, the temperature rise of hotspot at the nano-constricted region should reach as high as $\Delta T_L \sim 500 \text{ K}$ (i.e., $T_L \sim 800 \text{ K}$), which is strictly ruled out by the infrared thermography imaging. The far-field thermography images (similar to the one shown in Fig. 2 of Weng et al., Nano Lett. **18**, 4220(2018)) are taken but the result shows that the lattice temperature is not elevated discernibly ($\Delta T_L < 20 \text{ K}$; *Otherwise the hotspot at the channel center should be unambiguously observed in the thermography image taking account of its spatial and temperature*

resolution) in a $120\ \mu\text{m} \times 120\ \mu\text{m}$ area by application of $P=25\ \text{mW}$ to a similar Si device with a constricted channel. In brief, we are convinced that the different Seebeck coefficient in the constriction cannot explain quantitatively the large thermoelectric signal as shown in Figs. 2d-h and Figs. 3b,e.

Following Reviewer's suggestion, we have added the description "...Therefore the Seebeck coefficient change cannot play a dominant role for the observed thermoelectric results..." when we describe Fig. 2d-h and Figs. 3 b,e. We also replot Figs 3 c,f with additional data for smaller bias ($V_b=3\text{V}$) which continue to follow the cubic dependence on the current ($\Delta T_{\text{Peltier}} \propto I^3$), as also shown in the following Fig. R2. Data with even smaller bias voltage is restricted by the low signal-to-noise ratio since Peltier signal decreases more rapidly than Joule heating with decreasing the bias ($\Delta T_{\text{Joule}} \propto I^2$) in our experiments.

Fig.R2 Replot of Fig.3 c,f with additional data at smaller bias (3V).

3. Significantly, in order to rule out the local modification of the Seebeck coefficient in the junction, the scanning thermal gate microscopy (STGM) can be used [Harzheim et al, 2D Materials 2020, 7 (4), 041004. DOI: 10.1088/2053-1583/aba333.] where the hot tip creates the heat gradient and the resulting potential is measured. Such study should be pretty straightforward in the setup.

Reply: We thank the Reviewer for this very good suggestion. We have followed Harzheim et al's work (Ref.13 and newly added Ref. 47) and used hot tip to create heat gradient in our sample and record the potential change with scanning the tip position on our sample. As shown in Fig. R3, no discernible signal can be detected at the nano-constricted region (two dimensional maps in Figs. R3 h,i and line profile in Fig. R3j) while prominent signal is visible at the edge in contact with metal electrodes (Figs. R3c,d). The red and blue signal appearing at mesa edges in (h) and (i) is due to symmetry-breaking in hot-tip-induced electron diffusion. The absence of signal at the nano-constricted center in Fig. R3(h) suggests that the geometrical confinement plays little role in our experiments, while the absence of signal in (i) implies applying current does not induce appreciably Seebeck coefficient change even for the narrowest region with large electrical current density. Following Reviewer's suggestion, we have added Supplementary Note 9 which includes Fig. R3 as Fig. S12.

Fig. R3 (a) Schematic measurement setup of scanning thermal gate microscopy with heated tip and (b) microscopic image of sample with I, II, III labelling the different regions. The line profiles of measured sample thermos-voltages at different tip heating bias (4V, 5V, 6V) in region I (c) and region III (d) crossing the edge of metal electrodes. The sample thermos-voltage signal is proportional to the tip temperature rise with negative polarity in region I (e) and positive in region III (f). The temperature rise of the tip is calibrated and found to be proportional to the square of tip bias as shown in (g). In the nano-constricted region II, no apparent sample thermos-voltage can be obtained for both without (h) and with +10V (i) source-drain bias and the absence of the sample thermos-voltage signal is also seen in the line profiles (g). Artificial signal appears at mesa edges due to symmetry-breaking in hot-tip-induced electron diffusion (red and blue in (h) and (i)).

4. The description of the Scanning Thermal Microscopy to map lattice temperature is very inadequate. The design of the local temperature measurements, that are strongly dependent on the tip-surface contact thermal resistance, the verification of the measured surface temperatures had to be done via one of the existing methods for active or passive devices [see e.g. Menges et al, Nature Comm 2016, 7, 10874, Article. DOI: 10.1038/ncomms10874] and [Evangelini et al, Carbon 2021, 178, 632-639. DOI: <https://doi.org/10.1016/j.carbon.2020.12.012>].

Reply: We thank the Reviewer for this comment which makes us realize that description about the experimental details is insufficient. Indeed, acquiring the actual temperature from the original SThM

signals is very hard in actual situations, many factors would impact the temperature measurements [Refs.37,52], such as the thermal resistance of SThM probe, tip-surface contact and cantilever, even the feedback laser focus on the cantilever can also greatly influence the output signals. This makes the careful calibration necessary for us to analyze the data more quantitatively. In our experiments, a thermocouple-type probe technology (based on the Seebeck effect) was adopted for measuring the lattice temperature, as previously used in Refs. 14 and 52. When the thermocouple tip contacts with the sample surface, the temperature difference ΔT is established between tip apex and end of the cantilever (heat sink), and consequently causing the thermovoltage ΔV according to the $\Delta V = S_{tip}\Delta T$ where S_{tip} is the Seebeck coefficient of thermocouple. For accurate measurement of surface temperature, the sensitivity of the thermal probe ($\mu\text{V}/\text{K}$) is required which is determined by calibrating it against a range of known temperatures (Fig. R4). For this purpose, the thermocouple probe (tip) was brought into contact with the silicon flake pasted tightly onto the micro heater that is integrated with a standard K-type thermocouple, as depicted in the inset of Fig. R4. With this method, different factors affecting the real sample measurement may also occur on this reference sample with known temperature. Hence, the influence from these factors (e.g., the tip-surface contact thermal resistance) to the measured temperature value on real samples can be largely suppressed. On the reference sample, the sensitivity of the thermocouple probe is measured carefully (dots in Fig. R4) and extracted to be about $13 \mu\text{V}/\text{K}$ as shown by the slope of the linear fitting of experimental data. This sensitivity value agrees reasonably with the specification of the tip and also Ref. 52.

Fig. R4 calibration experiments by measuring thermos-voltage of the thermocouple tip in close contact with silicon pasted tightly onto the micro heater with known temperature. The sensitivity of the probes is measured and extracted to be about $13 \mu\text{V}/\text{K}$. Inset: Schematic drawing of the calibration setup.

Beside the above careful calibration, we have also estimated the possible temperature rise in our sample. Considering the real geometry of our device as schematically shown in Fig.R5, Joule heat is generated approximately in a limited area $\sim 1 \mu\text{m}^2$ of the device. Since the device is epitaxially grown on the Si single crystal substrate, the generated heat, in turn, spreads quasi-isotropically in 2π solid angle in the substrate without being hindered at the heterostructure interfaces as shown below. The heat flow thereby dilutes as $1/r^2$. The temperature rise at distance r from the hot spot is whereby derived to be $\Delta T_L = P/2\pi r K_L$ with P the Joule heating power and K the lattice thermal

conductivity of GaAs. Hence, we obtain $\Delta T_L \sim 20$ K at $r = 5 \mu\text{m}$ for $P \sim 25$ mW and $K_L \sim 40$ W/m·K (Si), which is reasonably consistent with the order of measured data. (The overall heating of the Si substrate is far smaller because the total volume is much larger.) Our estimation in the above is also consistent with a more involved simulation calculation (L. Yang et al., EPL **128**, 7001 (2019)). All these reassure the non-equilibrium condition ($T_L \ll T_e$) indeed taking place in our work.

Fig. R5 simple structure of heat diffusion for estimation of ΔT_L

Following Reviewer's suggestion, we have added more details of SThM measurements in Method section and in newly added Supplementary Note 9. Besides, we also mention that the measurement of T_L with SThM is not affected by the hot electron subsystem because the tip touches directly the lattice but not electron subsystem which has much smaller heat capacity by several orders of magnitude than lattice.

5. The experimental setup details of the SThM and error analysis are also very insufficient. Without these, the validity of the results cannot be verified.

Reply: We thank Reviewer for this comment. In our revised manuscript, we have included the experimental setup details of the SThM measurements. For error analysis, the typical line profiles of ΔT_{Joule} and $\Delta T_{Peltier}$ are shown in Fig. R6. From this figure, the typical error for Joule heat is about ~ 35 mK and for Peltier data, ~ 20 mK (indicated by 24 lines in Fig. R5(b)). Accordingly, the error bar for $\Delta T_{Peltier}$ has been added in Fig. 3c.

Fig. R6 typical line profiles of ΔT_{Joule} (a) and $\Delta T_{Peltier}$ with 24 repeated lines (b)

We mention that considering the facts that the excellent reproducibility, careful calibration in SThM, quantitative evaluation of reasonable temperature rise and observed the cubic power law dependence of $\Delta T_{Peltier}$ on current, all these show excellent consistency with our non-equilibrium thermoelectric scenario, making our results convincingly valid.

6. While the description of the SNoiM is reasonable in the references 35,36 a description of this method with the experimental setup used has to be presented in the SI and summarised in the Methods section.

Reply: We thank Reviewer for this comment and suggestion. In our revised manuscript, we have added the details of SNoiM setup in Supplementary Note 4 and summarized in the Methods section, so that readers can understand the necessary information about SNoiM without looking up Refs.35,36. In particular, we add in Method section "...SNoiM exclusively senses evanescent radiation localized on the material surface, but does not sense the familiar THz photon emissions such as those due to the blackbody radiation, externally induced coherent electron motion, and the one-particle radiative transition between the initial and the final states. This is because all those photon emissions do not yield evanescent field on the material surface. Detected with SNoiM is the charge/current fluctuation that generates intense evanescent waves but cancels out in the region away from the surface. In this work it is the *hot-electron shot noise*, the intensity of which is most simply characterized by the effective electron temperature T_e . Absolute values of T_e are derived from the signal intensity without using any adjustable parameter...". And in Supplementary Note 4, we provide technical details of SNoiM measurements.

Minor points.

1. The English of the manuscript, while reasonable, has to be carefully revised by the native speaker.

Reply: We thank Reviewer for this comment and suggestion and we have asked an experienced researcher to carefully proof-check and improve our manuscript. Hope that it is now reaching a satisfactory level.

2. Error bars has to be provided or indicated in the graphs.

Reply: We thank Reviewer for this suggestion and we have added the error bar for both SThM (Fig. 3c) and SNoiM data (inset of Fig. 4b).

3. The heat transport Eq. 1 relates to both electron and lattice that has to be corrected.

Reply: We thank Reviewer for this suggestion and we have made the minor correction to Eq. (1), as suggested by Reviewer. Also, we have added more details of the derivation of Eq. (1) in Supplementary Note 8

In summary, we are convinced that, in the revised version of the manuscript, our interpretation and conclusions are convincing and clear enough. We hope that Reviewer is satisfied with our new version of the manuscript, along with our response in the above.

Reviewer #2 (Remarks to the Author):

The authors report the direct observation of the charge-to-heat current conversion in a Si-based nano-constriction by means of SNoiM and SThM techniques. Thermoelectric cooling/heating signals were observed to appear in the nano-constriction despite the absence of junctions comprising different materials. The authors attribute the observed behavior to the non-equilibrium between electron temperature T_e and lattice temperature T_L . Active thermal management for Si-based electronic devices is of crucial importance, and this work provides a fundamental step to realize such a functionality. The manuscript is well written and organized. However, for further consideration, the following issues must be addressed.

Reply: We thank the Reviewer very much for his/her careful review and for the very positive comments to our work. We also appreciate the additional comments which are very much valuable for us to improve our manuscript. We have addressed each point of the report and tried our best to improve our manuscript accordingly.

1) The observed phenomenon should not be called the Peltier effect but is rather closer to the Thomson effect. The temperature change observed in the nano-constriction device is caused by ∇S , which is indeed the origin of the Thomson effect. I understand that the observed phenomenon due to the difference between T_e and T_L is different from the conventional Thomson effect. Nevertheless, the non-linear thermoelectric phenomenon due to ∇S should be phenomenologically categorized into the Thomson effect, not the Peltier effect. The authors should reconsider and discuss this point.

Reply: We thank Reviewer very much for this comment which is fundamentally important. We agree that what we observed is neither the conventional Peltier effect nor the conventional Thomson effect. We also agree that phenomenologically the phenomena we observed appears more like Thomson effect since it arises from non-uniform temperature distribution ($\nabla T \neq 0$). After careful reconsideration, we address this point in the following steps. First, we provide the detailed derivation of the analytical formula which dominates our experimental observation and meanwhile compare with the conventional Peltier and Thomson terms. Second, we discuss about the terminology and categorization. Third, based on the above consideration and following Reviewer's suggestion, we explain how we improve our manuscript in order to deliver the transparent key messages to readers while avoid misleading or confusion.

(1) Derivation of basic heat transport equation

The heat flow (\vec{q}) in current-carrying devices (\vec{j}) can be written as,

$$\vec{q} = ST\vec{j} - \kappa \cdot \nabla T \quad (\text{R1})$$

Where the first term on right side is the thermoelectric heat (with ST being the Peltier coefficient Π according to Thomson relation, i.e., $\Pi = ST$) and the second is the local heat dissipation through thermal conductance κ . Under steady-state condition with time-independent temperature distribution,

$$-\nabla \cdot \vec{q} + \dot{q} = 0 \quad (\text{R2})$$

Where \dot{q} is the local heat generation and is determined by the Joule heating noting that electric field \vec{E} is the only external driving force. Namely,

$$\dot{q} = \vec{E} \cdot \vec{j} = \frac{j^2}{\sigma} \quad (\text{R3})$$

Combing Eqs. R1-R3 and noting that $\vec{\nabla} \cdot \vec{j} = 0$, we obtain,

$$\frac{j^2}{\sigma} + \vec{\nabla} \cdot (\kappa \vec{\nabla} T) - \vec{\nabla}(ST) \cdot \vec{j} = 0 \quad (\text{R4})$$

This equation applies for the current-carrying (non-uniform) conductors assuming quasi-thermal-equilibrium condition, i.e., $T_e \cong T_L = T$. Same formulism has been used previously in current-induced thermoelectric transport, for example, in Refs. [24, 40].

In our present work, above equations have to be renewed by distinguishing and including both T_e and T_L in order to take into account contributions from both electron and lattice.

- (i) The thermal conductance term should be changed to the sum of electron and lattice part, i.e., $\vec{\nabla} \cdot (\kappa \vec{\nabla} T) \rightarrow \sum_{i=e,L} \vec{\nabla} \cdot (\kappa_i \vec{\nabla} T_i)$, under the rational assumption that contributions of electron and lattice to the total thermal conduction are separable, as been demonstrated in our previous work (L. Yang et al., EPL **128**, 7001 (2019)).
- (ii) The temperature in the thermoelectric heat term ($-\vec{\nabla}(ST) \cdot \vec{j}$) should be replaced with T_e rather than T_L . Here, the non-equilibrium Peltier coefficient $\Pi_{\text{nonequilibrium}} = ST_e$ is consistent with previous theoretical derivation [Ref. 33] for non-equilibrium condition which correlates $\Pi_{\text{nonequilibrium}}$ with T_e instead of T_L , and has been used in theoretical works such as Eq.(77) in *Local equilibrium and off-equilibrium thermoelectric effects in silicon semiconductors*, JAP 110, 093706 (2011). This formulism can be safely used because the electron distribution under the nonequilibrium condition can be correctly described with the effective electron temperature T_e together with a quasi-Fermi level.[Refs.24,33]

Eventually, Eq. R4 is changed to,

$$\frac{j^2}{\sigma} + \sum_{i=e,L} \vec{\nabla} \cdot (\kappa_i \vec{\nabla} T_i) - \vec{\nabla}(ST_e) \cdot \vec{j} = 0 \quad (\text{R5})$$

Which is Eq.(1) in the Discussion part of our manuscript. In our experiments, the electrothermal (Joule heating) signal comes from the first term $\frac{j^2}{\sigma}$, and the measured thermoelectric signal is essentially the thermoelectric heat term, i.e.,

$$\dot{Q}_{\text{measure}} = -\vec{\nabla}(ST_e) \cdot \vec{j} \quad (\text{R6})$$

Which can be decomposed into two separated terms, i.e.,

$$\dot{Q}_{\text{measure}} = -S\vec{\nabla}T_e \cdot \vec{j} - T_e\vec{\nabla}S \cdot \vec{j} \quad (\text{R7})$$

The first term on the right side of Eq.(R7) is closely related to Seebeck effect since $-S\vec{\nabla}T_e \cdot \vec{j} = \vec{E}_{\text{diff}} \cdot \vec{j}$ and it physically means absorbed or generated heat by electrical current in order to overcome or comply with the Seebeck-effect-induced electric field (\vec{E}_{diff}). Quantitatively, our simulation suggests that this term dominates the observed signal in our experiments and, phenomenologically, this term indeed can be termed as Thomson effect although the term does not contain Thomson coefficient ($\mu \equiv T_L \frac{dS}{dT_L}$). The second term can be approximated as $-T_e\vec{\nabla}S \cdot \vec{j} \approx$

$-\frac{T_e}{T_L} \left(T_L \frac{dS}{dT_L} \right) \vec{\nabla}T_L \cdot \vec{j} = -\frac{T_e}{T_L} \mu \vec{\nabla}T_L \cdot \vec{j} = \frac{T_e}{T_L} Q_{\text{conventional Thomson}}$. For easy comparison, we rewrite

the conventional Pelter and Thomson,

$$\dot{Q}_{\text{conventional Peltier}} = (\Pi_A - \Pi_B)\vec{j} \quad (\text{R8})$$

$$\dot{Q}_{\text{conventional Thomson}} = -\mu \vec{\nabla} T_L \cdot \vec{j} = -\left(T_L \frac{dS}{dT_L}\right) \vec{\nabla} T_L \cdot \vec{j} \quad (\text{R9})$$

From Eqs. R8 and R9, it can be seen that the conventional Peltier effect occurs at the interface of two dissimilar materials $\Pi_A \neq \Pi_B$, while the conventional Thomson comes from the temperature-dependence of the Seebeck coefficient $\mu \neq 0$.

Therefore, it is clear that our observation ($-\vec{\nabla}(ST_e) \cdot \vec{j}$ in Eq.R6), dominated by ($-S\vec{\nabla}T_e \cdot \vec{j}$ in Eq.R7), differs from both conventional Peltier and Thomson effects (Eqs. R8, R9).

Noting that our work is the first experimental demonstration of the nonequilibrium thermoelectric effect in non-uniform Si conductors, we are also aware that previous theoretical work used both Peltier and Thomson terminology. For instance, the term of ‘‘Peltier effect’’ was used in Refs.[23, 25, 33] and the term of ‘‘Thomson effect’’ was used in Ref. [24] and JAP **114**, 033704 (2013). Besides, people notice that all three thermoelectric effects (Seebeck, Peltier, Thomson) are correlated and combined to $ST\vec{j}$ (Eqs. R1 and R6): the term of interest ‘‘which gives the thermoelectric effects: directly the Peltier cooling but indirectly the Seebeck effect and Thomson effect’’, as written in Energy **56**, 61 (2013) and references therein.

Taking account of above situation and following Reviewer’s comment, we provide the above expressions in Supplementary Note 8 so that the physical essence of the observed phenomena is delivered transparently to readers, meanwhile we use mainly more general term ‘‘(nonequilibrium) thermoelectric effect’’ including the modified title. Also, we explicitly explain that ‘‘Phenomenologically the observed signal appears more like Thomson effect’’ although nonequilibrium Peltier coefficient is used for simulation.

2) The quantitative accuracy of the T_e measurements using SNoiM should be demonstrated. For example, at local thermal equilibrium conditions, T_e should be the same as T_L . Thus, SNoiM data should exhibit T_e distributions comparable to T_L distribution detected by SThM. Such an experimental demonstration under local thermal equilibrium conditions should be provided as a control experiment.

Reply: The precision of T_e in the measurement of SNoiM is within $\Delta T_e = \pm 100$ K in the region outside mesa (without conduction electrons) and $\Delta T_e/T_e = \pm 7\%$ in the region of mesa (with conduction electrons).

Measurement of thermal equilibrium system with SThM and SNoiM: The quantitative reliability of our SThM measurements is also shown in the figure below (Fig. R7), which displays results of additional experiments made on a Joule-heated narrow metal wire (20 nm-thick NiCr) deposited on a SiO₂/Si substrate (a). (The measurements are similar to the earlier study of SNoiM; Q.Weng et al., *Near-Field Radiative Nanothermal Imaging of Nonuniform Joule Heating in Narrow Metal Wires*, Nano Lett. 18, 4220 (2019)). In the measurements on metals, the resolution and the quantitative accuracy can be directly confirmed because (i) the temperature changes sharply at the edge of the heated metal wire and (ii) the conduction electrons are in quasi equilibrium with the lattice so that quantitative comparison with simulation calculation is possible ((b),(e); L.Yang et al., *Simulation of temperature profile for the electron and the lattice systems in laterally structured layered conductors*, EPL 128, 7001 (2019)). A hot-spot (b, c, d), showing up along the inner edge of the bended corner of the wire (currents $I=1\sim 8$ mA) due to local Joule heating caused by the current crowding effect, provides a convenient target for the test. A spatial resolution better than 100 nm is demonstrated by the sharp step-wise increase of the SThM signal at the inner-edge of the

wire (d). (Edge related artifact signal is also seen at the outer edge of the wire.) The 2D-images of simulation (b), SNoiM (c; T_e), and SThM (d; T_L) are similar to one another. Especially, (e) shows that the amplitude ΔT_L derived from SThM is substantially accounted for by the theoretically expected values given by the simulation calculation, confirming the validity of our SThM measurements in absolute terms.

Fig. R7 SNoiM and SThM measurements on a Joule-heated narrow metal wire (20 nm-thick NiCr) deposited on a SiO₂/Si substrate

3) The authors should describe how to determine the phase offset in Fig. 2. In the nano-constriction used in this study, the temporal response of the temperature change is affected not only by the device design but also the heat transfer to substrates. Thus, the temperature change should have a delay from the applied charge current, leading to non-zero phase values. However, in the data in Fig. 2, the thermoelectric signals have only 0 degree and 180 degrees without any time delay.

Reply: Thank reviewer for the comment. We found that the typical response time in our samples is about 10 ns-10 μs. The measurement in this work is typically at 413 Hz, or ~2.5 ms which is more than 3 orders longer than the typical thermal response time of our samples. Therefore, no apparent phase delay was seen in our experimental data. Similar data has also been reported, for example, in Ref.[37]

4) What is the periodic pattern on the right side of the phase image in Fig. 2g?

Reply: Sorry for the confusing phase image feature in our original manuscript. It was an experimental artifact which may be caused by occasional noise. We have repeated the measurement which shows no periodic pattern and confirmed the reliability of our measurements.

5) Related to the comment 3), the sentence "In principle, the Peltier signals are obtained by multiplying lock-in amplitude R and phase phi" (page 5) is not correct. The thermoelectric signals detected by lock-in techniques are often discussed in terms of $R \cdot \cos \phi$, which includes both the magnitude and sign information. However, this factor can be used only when the time delay due to thermal diffusion is negligibly small. Since such a time delay cannot be neglected in the present device, the above sentence is not valid (and "cosine" is missing).

Reply: We thank Reviewer very much for his/her careful review and kind suggestion. We have made the correction.

6) The combination of SThM and lock-in detection is one of the key techniques in this study. Thus, the procedures of the measurement conditions should be described in more detail. In the case of the well-known lock-in thermography, the lock-in frequency must be much smaller than the frame rate of the infrared camera. In a similar manner, in the lock-in SThM, does the lock-in frequency need to be much smaller than the cantilever scanning frequency? Conversely, can the lock-in frequency be much larger than the scanning frequency? I could not find the information on the scanning frequency in the SThM measurements.

Reply: We thank Reviewer for this comment and question. The typical modulation frequency in our SThM experiments is 413 Hz, which is sufficiently lower than the cantilever resonance frequency (tens of kilohertz). Meanwhile, it is quick enough for the data acquisition at each point during the imaging with line scan mode. In our revised manuscript, we have added more information about the lock-in SThM technique.

7) In the experimental results reported here, the thermoelectric cooling is much smaller than Joule heating. However, as demonstrated in Ref. 38 and Phys. Rev. Applied 11, 034022 (2019), the spatial distribution and temporal response of the temperature change induced by dipolar heat sources (a symmetric pair of heating and cooling sources due to thermoelectric effects) are different from those by single heat sources, such as Joule heating. Thus, in principle, the relative contribution of the thermoelectric effects can be larger by changing the design of the nano-constriction. This point should be discussed.

Reply: We thank Reviewer very much for making this inspiring comment and suggestion. The nonequilibrium thermoelectric effect observed in this work has cubic power dependence on the current and it relies on the Joule-heating-induced temperature gradient. It is therefore small at this moment and becomes more prominent for high-electric-field. On the other hand, a wide range of different methods may be considered to further improve, such as laser-spot-induced temperature gradient, simultaneous active cooling/heating source, intervalley-assisted energy transport, spin-Peltier effect and electron hydrodynamics etc. In our manuscript, we cite Phys. Rev. Applied 11, 034022 (2019) as Ref. 54 and add the discussion in the above.

8) In the thermoelectric measurements, the authors applied a square-modulated AC charge current to the device. Thus, when extracting the first harmonic response, the factor $4/\pi$, i.e., the amplitude difference between the square wave and first harmonic sinusoidal wave, should be taken into account. Is this factor included in the analysis?

Reply: We thank Reviewer very much for this comment and question. Yes, this factor should be taken into account. We have calibrated this factor (~ 1.2 in our experiments) and included this factor into our revised manuscript.

9) In Supplementary Note 4, "marked by the blue dashed line in Fig. S4(b)" should be replaced with "marked by the black dashed line in Fig. S4(b)".

Reply: We thank Reviewer very much for his/her careful review and kind suggestion. We have made the correction.

Reviewer #3 (Remarks to the Author):

In their paper Xue et al. show that a local non-equilibrium state where the electronic system is hotter than the lattice might alter the local Peltier coefficient. The experiments and their interpretation are very interesting. I would in principle suggest this work for publication, but I have a few small questions.

Reply: We thank the Reviewer very much for evaluating our experiments and interpretation to be very interesting and for suggesting our work to be published. We are grateful to the comments which are very much valuable for us to improve our manuscript. We have addressed each point of the report and improved our manuscript accordingly.

I have some fundamental understanding issue. In my opinion, the Peltier effect (as well as the Seebeck effect) are happening in the electronic system. Meaning: a difference in electron temperature drives an electrical current (Seebeck effect) and a charge current drives a heat current via energetic electrons (Peltier effect). The reason that the lattice temperature is often used in literature is due to the fact that thermalization between electronic and lattice systems is usually very fast (high electron-phonon interaction strength at room temperature). Do you exclude that Peltier cooling should occur in the electronic system as well? The way I understand your Figure 4a is that you say that the Peltier effect is a consequence of heat transfer between the lattice and the electronic system, which can be either positive (cooling) or negative (heating) depending on the gradient of T_e . But wouldn't you expect already a heating/cooling of the electronic system only due to the Peltier effect (=electron-hole asymmetry of the DOS at the Fermi energy, asymmetry of heat transfer, net heat flow, ...). And would you be able to measure this using your SNoiM method (similar to what you do with SThM, by applying an AC signal to the device and by locking to this signal)?

Reply: We thank the reviewer for this very important question. The Peltier effect in conventional quasi-equilibrium system does not distinguish whether it comes within the electron subsystem or from the electron-lattice interaction. In our far-from-equilibrium case, however, there are two possible mechanisms as Reviewer suggested. If we assume that electron subsystem and lattice subsystem are *fully separable* during the electron thermoelectric transport, then Peltier effect can occur within the electron subsystem. In this case, T_e -profile should be imprinted into T_L -profile through thereafter electron-phonon interaction. Therefore, T_e and T_L should have same spatial distributions except their different peak amplitudes. In our experiments, however, T_e shows same peaks for $V_b=0/+10V$ pulsed bias and for $V_b=0/-10V$, as shown in Fig. R8(a); while T_L shows deviated peaks ($\Delta x \sim 250$ nm) for the same positive and negative unipolar bias conditions (Fig. R8(c), replotted from Fig. S4 in Supplementary Information). As Reviewer recommended, we also tried SNoiM measurement with bipolar AC bias and the result shows no discernible electron-Peltier signal (Fig. R8(c)). This is remarkably different from the result for lattice-Peltier signal as shown in Fig. R8(d) (replotted from Fig. S4). We therefore conclude that Peltier effect within the electron subsystem cannot be dominant in our device while electron-phonon scattering plays already the decisively important role in the observed Peltier effect. In this process, gradient of T_e with respect to the current direction determines the positive/negative Peltier effect (cooling/heating) to be observed from the lattice subsystem, as discussed in main text of our manuscript.

Following Reviewer's comment, we have added in our revised manuscript the discussion about the absence of Peltier effect within electron subsystem and mention the SNoiM measurement with bipolar bias (Fig. S4 e, f).

Fig. R8 Comparison of SNoiM (a and b) and SThM (c and d) experiments with unipolar 0~+10 V (red) or 0~+10 V (blue) (a and c) and bipolar -10 V ~+10 V (b and d) pulsed bias V_b .

I am also not sure if I fully understand your explanation of the local variation of the Peltier coefficient. Is it correct that you say that the local change in Peltier coefficient is in your case a non-equilibrium effect which is purely caused by a raise of the electronic temperature ($T_e \gg T_L$). This local variation in electron temperature gives you a spatial variation of the quasi-Fermi level and thus a variation of the Peltier coefficient?

Reply: We thank the reviewer for this question. Yes, Peltier coefficient depends on the Seebeck coefficient and the local temperature, according to Thomson relation. The spatial variation of the quasi-Fermi level in nano-constricted region of our device leads to a variation of the Peltier coefficient.

So, out of curiosity, if you present evidence that there is a big difference between the electron and the lattice temperature when Joule-heating your junctions: 1) is this an effect that you would only see for high electric fields? And is there a threshold voltage where this occurs? 2) Since $T_e \gg T_L$ is a non-equilibrium situation, on what time-scale does the system relax/thermalize ($T_e = T_L$)? My feeling tells me that electron-phonon interactions at room temperature are rather strong, so I would guess that this thermalization happens in the nano-second range. Is this within your experimental range?

Reply: We thank the reviewer for the questions and comments.

1) The effect indeed becomes prominent for high electric fields. In case of low electric fields, it is hardly to be discernible because $\Delta T_{\text{Peltier}} \propto I^3$ and the amplitude decreases faster than Joule contribution ($\propto I^2$) when I decreases. Experimentally, we cannot identify the threshold voltage but the power-law dependence on current (symbols in Fig. 3c).

2) The typical scattering time for electrons in our device at room temperature is on the order of ps [JAP **52**, 6713 (1981); Ref. 53, Nat. Comm., **8** 15177 (2017) etc.] and the thermalization between electron and lattice subsystems can be very fast, for example, in nanosecond or shorter, if the electrons are pumped to a high temperature (T_e) with a transient pulse (as in Ref. [53]). However, in our electrical measurement, the device works in a *steady-state*, meaning that the electron subsystem is constantly acquiring energy from external electric field and meanwhile losing energy to lattice subsystems through electron-phonon interaction. The dynamic balance between energy gain and loss of electron subsystems leads to an elevated electron temperature being much higher than the lattice ($T_e \gg T_L$). (Note that the modulation of source-drain bias ($f_{\text{exc}} \sim 413$ Hz) is much slower than the typical transient thermalization time and the modulation is simply for the purpose of subtracting the thermal background and increase the signal-to-noise ratio by lock-in technique in our measurements.) Beside the constant energy input to the electron subsystem, the steady-state with nonequilibrium electron and lattice temperatures ($T_e \gg T_L$) also arises from the fact that the heat capacity of electron subsystem is intrinsically small, being several orders of magnitude less than that of lattice. We take all these aspects into consideration in our previous work [L. Yang et al., EPL **128**, 7001 (2019)] and in the numerical simulation of present work as shown in Supplementary Note 7 (Table S1 and related text).

Following Reviewer's question and comments, we have tried to improve the description in our manuscript and add explicitly the *steady-state* condition of our device. In Method section, we also mention that "SNoiM is so far the only instrument that visualizes hot electrons in the steady-state transport condition..." to emphasize the application of SNoiM for steady-state measurement.

Minor comment: Figure 2 caption: scale bar 400nm not um.

Reply: We thank the reviewer very much for the careful review and sorry for this mistake. In the revised manuscript we have corrected this error.

REVIEWERS' COMMENTS

Reviewer #1 (Remarks to the Author):

The revised manuscript generally addresses all the key questions from all three editors and can be recommended for publication.

Reviewer #3 (Remarks to the Author):

All comments have been addressed. I suggest publication.

Reply to Reviewer #1 and Reviewer #3:

Reviewer #1 (Remarks to the Author):

The revised manuscript generally addresses all the key questions from all three editors and can be recommended for publication.

Reviewer #3 (Remarks to the Author):

All comments have been addressed. I suggest publication.

Reply: We thank the Reviewers very much for the comments and for supporting our work to be published.